# Annual replication is essential in evaluating the response of the soil microbiome to the genetic modification of maize in different biogeographical regions

**Márton Szoboszlay[1], Astrid Näther[1], Ewen Mullins[2], Christoph C. Tebbe[1]***

**1** Thünen Institute of Biodiversity, Federal Research Institute for Rural Areas, Forestry and Fisheries, Braunschweig, Germany, **2** Teagasc, Agriculture and Food Development Authority, Dept. Crop Science, Oak Park, Carlow, Ireland

\* christoph.tebbe@thuenen.de

**Data Availability Statement:** All relevant data, except for DNA sequences, are within the manuscript and its Supporting Information files.

## Abstract

The importance of geographic location and annual variation on the detection of differences in the rhizomicrobiome caused by the genetic modification of maize (Bt-maize, event MON810) was evaluated at experimental field sites across Europe including Sweden, Denmark, Slovakia and Spain. DNA of the rhizomicrobiome was collected at the maize flowering stage in three consecutive years and analyzed for the abundance and diversity of PCR-amplified structural genes of *Bacteria*, *Archaea* and *Fungi*, and functional genes for bacterial nitrite reductases (*nirS*, *nirK*). The *nirK* genes were always more abundant than *nirS*. Maize MON810 did not significantly alter the abundance of any microbial genetic marker, except for sporadically detected differences at individual sites and years. In contrast, annual variation between sites was often significant and variable depending on the targeted markers. Distinct, site-specific microbial communities were detected but the sites in Denmark and Sweden were similar to each other. A significant effect of the genetic modification of the plant on the community structure in the rhizosphere was detected among the *nirK* denitrifiers at the Slovakian site in only one year. However, most *nirK* sequences with opposite response were from the same or related source organisms suggesting that the transient differences in community structure did not translate to the functional level. Our results show a lack of effect of the genetic modification of maize on the rhizosphere microbiome that would be stable and consistent over multiple years. This demonstrates the importance of considering annual variability in assessing environmental effects of genetically modified crops.

## Introduction

The cultivation of genetically modified (GM) maize with resistance against agricultural pests has become common practice in many countries across the world [1]. Among the genetic modifications, the expression of insecticidal endotoxins which are naturally produced by the

DNA sequences are deposited at the European Nucleotide Archive (https://www.ebi.ac.uk/ena) under the accession number PRJEB33038.

**Funding:** The study was funded by the Seventh Framework Program of the EU, Grant agreement 289706 (AMIGA: Assessing and Monitoring the Impacts of Genetically modified plants on Agro-ecosystems.

bacterium *Bacillus thuringiensis* (Bt), is especially important. Bt-modifications can provide resistance against insects that are not easy to control by chemical pesticides [2]. Bt-maize event MON810 has been widely used in agriculture for several decades. It produces the Cry1Ab toxin which kills the European corn borer (*Ostrinia nubilalis*), a widely abundant lepidopteran pest. Nowadays, the event MON810 is present in a high diversity of maize varieties and thereby occurs in agroecosystems with contrasting climatic conditions [1]. In Europe, maize can be cultivated across the whole continent with the exception of larger parts of Scandinavia, and the use of Bt-maize MON810 in the European Union was authorized by the European Commission in 1998 [3]. However, overruling national regulations have limited the actual commercial cultivation of maize MON810 to only a few European countries, with the most widespread use in Spain [1].

For authorizing the cultivation of any genetically modified crop in the European Union, legislation requires that environmental risks are characterized. Guidance documents developed by the Panel of Genetically Modified Organisms of the European Food Safety Authority explain the procedure for environmental risk assessment (ERA) [4]. An important aspect of ERA is the consideration of GM-specific effects on non-target organisms (NTOs), as they may interact with the GM plants in agroecosystems or beyond in neighboring ecosystems. Their protection is required to preserve biodiversity and maintain ecosystem functions [5]. NTOs encompass insects, earthworms, nematodes and other faunal groups, but they also include soil microorganisms, the latter involved in biogeochemical cycling and other ecosystem services [6].

The most immediate interactions between plants and soil microorganisms occur in the rhizosphere, i.e. the soil influenced by roots and their exudates [7]. The energy rich carbon sources provided by plant roots support a diverse microbial community dominated by bacterial and fungal taxa [8]. The composition of these rhizomicrobiomes is variable. It depends on the particular soil with its resident microbial community, as well as on plant properties, climate and other environmental factors [9, 10]. A diverse rhizomicrobiome is generally assumed to support the growth of the plant by facilitating its access to nutrients [11, 12] and provide protection against soil-borne plant pathogens [13, 14]. Thus, maintaining a diverse rhizomicrobiome could be an option in support of an environmentally friendly crop production and a protection goal when considering the unintended environmental effects of a GM plant [15].

The most suitable technical approach to capture the diversity of the rhizomicrobiome is the analysis of genetic markers in DNA extracted from rhizosphere samples. Suitable targets for bacterial and archaeal diversity is the 16S rRNA gene, and for fungal diversity the 18S rRNA and 28S rRNA genes and ITS sequences [16–18]. While these genes allow the characterization of the structure and composition of the microbial community, functional genes may provide additional information on an ecosystem service. Among them, bacterial *nirK* and *nirS* genes encode alternative versions of the enzyme nitrite reductase which mediates a key activity in denitrification, a process highly relevant in the rhizosphere. Denitrification results in a loss of nitrogen delivered by fertilization thereby wasting resources in crop production [12, 19]. While a change in the abundance or diversity of such marker sequences does not necessarily directly translate to an adverse effect of a GM plant, it can serve as an indicator if plant signaling to soil is altered, making further consideration of potential environmental hazards necessary.

The rhizomicrobiome of maize MON810 in different genetic backgrounds and in relation to near-isogenic cultivars as comparators has already been studied at a number of field sites located in different climatic regions, including Portugal, Spain, Slovakia, the USA, Brazil or Germany [18, 20–25], and an effect of the genetic modification was either not detected or minor compared to the impact of other variables including field heterogeneities, plant age, or

annual weather conditions. While these studies all utilized one or several of the above mentioned genetic markers, the resolution by which the corresponding DNA sequences were retrieved and the depth to which they were analyzed varied substantially. Due to the potential of high-throughput sequencing utilizing the Illumina platform, it is now possible to characterize the structural and functional diversity of rhizomicrobiomes in more depth, also retrieving the less abundant but not necessarily less important microbial community members. To date, this technique has to our knowledge not yet been applied to analyze the rhizomicrobiome of maize MON810.

Once approved, the European Union (EU) allows the cultivation of a GM plant in all the different geographical zones of the continent, not necessarily asking for specific data from all particular zones. This initiated the EU project AMIGA (https://cordis.europa.eu/project/rcn/101406/factsheet/en) in which context this study took place.

The objective of this study was to evaluate with a highly sensitive high-throughput DNA sequencing approach the impact of field site locations in different geographical zones on the rhizosphere microbial community of Bt-maize MON810 in comparison to near isogenic control varieties. Contrasting results between geographical zones could challenge the current ERA requirements for cultivating GM maize in Europe. In accordance with the Regulation on Plant Protection Products [26], we distinguished in this study three geographic zones, i.e. northern (N), central (C) and southern (S), and collected rhizosphere samples at experimental sites in Sweden and Denmark (N), Slovakia (C), and Spain (S). The event MON810 was present in three different cultivars which were suitable for cultivation at these sites. Each site was studied with annual replication. Molecular markers to quantify the abundance and characterize the diversity of the microbial communities from DNA extracted from the rhizosphere included bacterial 16S rRNA genes, fungal ITS sequences and bacterial *nirK* and *nirS* genes.

## Materials and methods

### Field sites and sampling

This study included four field sites located in different countries: Denmark, Slovakia, Spain, and Sweden. No specific permission was required for soil and rhizosphere sampling at these sites in context of the collaboration within the EU research project AMIGA (for more details see Acknowledgment section). The field studies did not involve sampling of endangered or protected species.

At each site, 10 plots were sown with BT maize and 10 plots with a near-isogenic non-BT cultivar. The maize cultivars used are listed in Table 1. The plots were randomly assigned to the cultivars. They were 10 m × 10 m in size, placed 5 m apart from each other, and bordered by 5 m wide conventional maize strips.

The site in Denmark was located at the Aarhus University's research center in Flakkebjerg (55˚19′N, 11˚23′E, 35 m above sea level, a.s.l.). The soil was a Udoll with pH ($CaCl_2$) 6.3 ± 0.3, 1.33 ± 0.08% total organic C and 0.14 ± 0.02% total N content. The site in Slovakia was west of Borovce (48˚34′N, 17˚43′E, 181 m a.s.l.) located on soil classified as a Udoll with pH ($CaCl_2$) 6.2 ± 0.5, 1.25 ± 0.13% total C and 0.14 ± 0.02% total N content. The site in Spain was southeast from Seseña (40˚05′N, 3˚40′W, 495 m a.s.l.) on a Fluvent with pH ($CaCl_2$) 7.6 ± 0.1, 1.57 ± 0.19% C, and 0.12 ± 0.01% N. The site in Sweden was located northwest of Lund (55˚45′N 13˚2′E, 10 m a.s.l.). The soil was classified as an Ochrept with pH (CaCl2) 5.94 ± 0.38, 1.43 ± 0.17% C, and 0.15 ± 0.03% N.

Rhizosphere samples were collected during the flowering stage (BBCH65) in 2012, 2013, and 2014. The site in Denmark was not sampled in 2012 and the site in Spain was not sampled in 2014. Plants were extracted from the soil, and loosely attached soil was removed from the

**Table 1. Maize cultivars cultivated and analyzed at the different sites and years.**

|  |  | 2012 | 2013 | 2014 |
|---|---|---|---|---|
| **Denmark** | Bt cultivar | n.c. | DKC3872YG | DKC3872YG |
|  | non-Bt cultivar | n.c. | DKC3871 | DKC3871 |
| **Slovakia** | Bt cultivar | DKC3872YG | DKC3872YG | DKC3872YG |
|  | non- Bt cultivar | DKC3871 | DKC3871 | DKC3871 |
| **Spain** | Bt cultivar | DKC6451YG | DKC6451YG | n.c. |
|  | non- Bt cultivar | DKC6450 | DKC6450 | n.c. |
| **Sweden** | Bt cultivar | DKC4442YG | DKC3872YG | DKC3872YG |
|  | non- Bt cultivar | DKC440 | DKC3871 | DKC3871 |

n.c., not cultivated

roots by shaking. To collect soil particles and microbial cells adhering to the roots, 8 g (fresh weight) of root was washed in 30 ml sterile saline solution (0.85% w/v NaCl) in a 50 ml Falcon tube by mixing for 30 min at 4˚C with 10 rpm on a rotary shaker. The roots were then discarded and the solution was centrifuged for 30 min with 4,100 x $g$ at 4˚C to collect the soil particles and microbial cells. The pellets were resuspended in 1 ml sterile saline solution and transferred to 1.5 ml test tubes. After 5 min centrifugation with 4,100 x $g$ at 4˚C, the supernatant was discarded and the pellets were stored at -80˚C until DNA extraction.

## DNA extraction and quantitative PCR (qPCR)

DNA was extracted from the initially frozen pellets with FastDNA SPIN kit for soil (MP Biomedicals, Illkirch, France) with two bead beating steps at 6.5 m s$^{-1}$ lasting 45 s each on a FastPrep-24 instrument (MP Biomedicals, Eschwege, Germany). An additional washing of the binding matrix with 1 ml 5.5 M guanidine thiocyanate was included.

Copy numbers of bacterial and archaeal 16S rRNA genes, *nirK*, *nirS*, and fungal ITS sequences were determined by quantitative PCR. Reactions were run in a StepOnePlus Real Time PCR System (Life Technologies GmbH, Darmstadt, Germany) in duplicates: once with 2 µl of 10 x diluted, and once with 2 µl of 50 x diluted DNA-extract in 20 µl final volume. The sequences of primers and probes with references are listed in S1 Table. For the quantification of bacterial and archaeal 16S rRNA genes, Maxima Probe qPCR ROX Master Mix (Thermo Fisher Scientific, Epsom, UK) was used with 0.5 µM primers and 0.2 µM FAM-labeled *Taq*-Man probes. The temperature profile was 10 minutes of denaturation at 95˚C followed by 40 cycles of 95˚C for 15 s and 60˚C for 60 s. The *nirK*, *nirS*, and fungal ITS qPCRs were carried out in Maxima SYBRGreen/ROX qPCR Master Mix (Thermo Fisher Scientific) and, in case of *nirK* and *nirS*, with the addition of MgCl$_2$ solution to increase its concentration by 1 mM. The temperature profile was 10 minutes of denaturation at 95˚C followed by 40 cycles of 95˚C for 15 s, 52˚C for 30 s and 72˚C for 60 s for fungal ITS, and 10 minutes of denaturation at 95˚C followed by 40 cycles of 95˚C for 15 s, 63˚C for 30 s decreased by 1˚C in every cycle to 58˚C, and 72˚C for 60 s for *nirK* and *nirS*. The reactions were followed by melt curve analyses. Standard curves were obtained from 10-fold dilutions of pGEM-T (Promega, Mannheim, Germany) containing either the 16S rRNA gene of *Bacillus subtilis*, the 16S rRNA gene of *Methanobacterium oryza*, the ITS region of *Fusarium culmorum*, the *nirS* gene of *Pseudomonas stutzeri*, or the *nirK* gene of *Sinorhizobium meliloti*.

The amplification efficiencies were 92.4 ± 1.7% for bacterial 16S rRNA genes, 93.4 ± 2.5% for archaeal 16S rRNA genes, 88.1 ± 4.3% for fungal ITS, 86.8 ± 2.1% for *nirK*, and 91.6 ± 1.5% for *nirS*. The qPCR results were expressed as copies g$^{-1}$ root (fresh weight) and subjected to

log10 transformation to decrease the heterogeneity of variance. Samples from BT and non-BT maize from the same site and year were compared with t-tests in JMP 13.0.0 (SAS Institute). Differences between sites and years were analyzed with ANOVA and Tukey's HSD post hoc tests, and pairwise Pearson correlations between the qPCR datasets were calculated in JMP 13.0.0. The significance of the correlations was assessed by t-tests after applying Fisher's Z-transformation.

### Sequencing and data processing

The V4 region of the bacterial and archaeal 16S rRNA genes and a segment of the *nirK* gene was amplified by PCR and subjected to Illumina MiSeq sequencing according to the protocol of [27]. The primers used in the PCR contained the Illumina adaptor sequences, 8 nucleotide (nt) long index sequences, a 10 nt section to allow the appropriate annealing temperature during sequencing, 2 nt not complementary to the target sequence, and a target sequence-specific region which was S-D-Arch-0519-a-S-15 (CAGCMGCCGCGGTAA) and S-D-Bact-0785-a-A-21 (GACTACHVGGGTATCTAATCC) [28] in case of the 16S rRNA gene, and F1aCu (ATCATGG TSCTGCCGCG) and *nirK*1040R (GCCTCGATCAGRTTRTGGTT) [29] in case of *nirK*. Two 16S rRNA gene and two *nirK* PCR were performed from each sample with FastStart High Fidelity PCR System (Roche Diagnostics, Mannheim, Germany) in 50 µl reaction volume containing 1 µl DNA-extract, 0.4 µM of each primer, 200 µM of each dNTP, 5% dimethyl sulfoxide, 1.8 mM $MgCl_2$, and 2.5 U FastStart High Fidelity Enzyme Blend. The temperature profile started with initial denaturation at 95°C for 2 min followed by 35 cycles of 95°C for 30 s, 50°C for 30 s, and 72°C for 60 s, and finished with 72°C for 5 min. The products of the two replicate reactions were pooled, purified with HiYield PCR Clean-up & Gel-Extraction kit (SLG) and quantified with Quant-iT PicoGreen dsDNA assay (Invitrogen, Darmstadt, Germany). Libraries were prepared from equimolar amounts of the individual PCR products and sent to StarSEQ (Mainz, Germany) for sequencing. The *nirK* PCR products were sequenced in a single MiSeq run with 600 cycle V3 chemistry. The 16S rRNA gene PCR products were distributed between two MiSeq runs, the first with 500 cycle V2 chemistry containing the samples from 2012 and 2013, and the second with 600 cycle V3 chemistry for the samples from 2014.

Fungal ITS sequencing was carried out from the samples from 2012 and 2013. ITS1 sequences were PCR-amplified using primers with Illumina overhang adaptors and target specific regions ITS1F (CTTGGTCATTTAGAGGAAGTAA, [30] and ITS2 (GCTGCGTTCTTCATCG ATGC, [31]. Reactions were carried out with Kapa2G Robust PCR ReadyMix (Kapa Biosystems, Wilmington, Massachusetts) in 50 µl volume containing 0.2 µM of each primer and 10 ng template DNA. The temperature profile started with initial denaturation at 95°C for 3 min followed by 35 cycles of 94°C for 45 s, 62°C for 45 s, and 72°C for 60 s, and finished with 72°C for 7 min. PCR products were cleaned with Agencourt AMPure XP beads (Beckman Coulter, Brea, California). Illumina sequencing adapters and index sequences were added in a limited-cycle PCR with 5 µl purified PCR product and primers from the Nextera XT Index kit (Illumina). The temperature profile started with initial denaturation at 95°C for 3 min followed by 8 cycles of 95°C for 30 s, 55°C for 30 s, and 72°C for 30 s, and finished with 72°C for 5 min. The products were purified with Agencourt AMPure XP beads (Beckman Coulter Life Sciences, Krefeld, Germany). Libraries were prepared from equimolar amounts of the individual PCR products and sequenced at the DNA Sequencing Facility of Teagasc (Moorepark, Carlow, Ireland) in two MiSeq runs with 600 cycle V3 chemistry. Sequences obtained in this project are available at the European Nucleotide Archive (http://www.ebi.ac.uk/ena) under the accession number PRJEB33038.

Sequences were processed with dada2 [32] version 1.3.5 in R 3.3.3 (https://www.r-project. org/). Error models were constructed and sequence variants (SVs) were determined for the

individual MiSeq runs separately. In the 16S rRNA gene dataset, forward and reverse reads were trimmed at position 10 from the start and at positions 240 and 180 from the end, respectively, or at any position with a Q-score of 2. Reads with more than two expected errors or any ambiguous bases were discarded. Error models were constructed with default settings from data from a subset of the samples containing over $10^6$ sequences in total. SVs were identified based on the error models using the pool option. Chimeric sequences were identified with the removeBimeraDenovo function of the dada2 package and discarded. The taxonomical classifications of the SVs were determined based on the SILVA reference version 123 [33]. Only results with at least 70% bootstrap support were accepted. SVs longer than 275 nt, or identified as mitochondrial or chloroplast sequences, or not classified into *Bacteria* or *Archaea* were removed from the dataset.

In the case of the *nirK* sequences, forward and reverse reads were trimmed at position 10 from the start and at positions 295 and 250 from the end, respectively, or at any position with a Q-score of 2. Reads with over two expected errors or any ambiguous bases were discarded. Error models were constructed with the band size parameter of the dada algorithm set to 40, and using data from a subset of the samples containing over 880,000 sequences in total. SVs were identified based on the error models using the pool option. Chimeric sequences were identified with the removeBimeraDenovo function of the dada2 package and discarded. FrameBot [34] was used to translate the SVs to amino acid sequences, correct for frameshifts, and compare the resulting translated sequence variants (TSVs) to the *nirK* reference database at the FunGene repository [35] with a minimal length cutoff of 80 amino acid residues and identity cutoff of 0.8.

The fungal ITS sequences were first processed with cutadapt version 1.13 [36] to trim off the primer sequences from the beginning of the reads and the complement sequence of the opposite primer from the end of the reads. A 10% mismatch rate was allowed in the recognition of the primer sequences. Forward and reverse reads in which the forward or reverse primer, respectively, could not be identified were discarded. Reads were truncated at the first position with a Q-score of 10 or lower and those shorter than 90 nt or with more than two expected errors or any ambiguous bases were discarded. Error models were constructed using all sequences from the sequencing run with the band size parameter of the dada algorithm set to 32. Chimeric sequences were identified with the removeBimeraDenovo function of the dada2 package and discarded. The taxonomical classifications of the SVs were determined based on the UNITE reference version 7.1 [37] with a bootstrap support threshold of 70%. SVs not classified as Fungi were removed from the dataset.

## Statistical analysis of the sequencing data

Simpson's diversity index was calculated from the 16S rRNA gene and fungal ITS SVs and the *nirK* TSVs using the vegan package [38] in R. Samples from BT and non-BT maize from the same site and year were compared with t-tests in JMP 13.0.0. Differences between sites and years were analyzed in JMP 13.0.0 with ANOVA and Tukey's HSD post hoc tests.

Principal component analysis (PCA) and non-metric multidimensional scaling (NMDS) plots based on Euclidean distance were created using the vegan package in R. To decrease the sparsity of the dataset 16S rRNA gene SVs and *nirK* TSVs that didn't have at least 0.10% relative abundance, or fungal ITS SVs without at least 1.0% relative abundance, in at least one of the samples included in the ordination were removed. Centered log-ratio transformation was used to correct for compositional effects and differences in sequencing depth between the samples [39]. To allow the transformation, 0.1 was added to all elements of the data matrices.

ALDEx2 with 1,000 Monte Carlo samples and Welch's t-test [40] was used to identify SVs, TSVs, or taxa differentially abundant between the samples from BT and non-BT maize from

the same site and year. Only the 16S rRNA gene SVs and *nirK* TSVs that had at least 0.10% relative abundance, or fungal ITS SVs with at least 1.0% relative abundance, in any of the compared samples were included. To obtain sequence counts of taxa, SVs were merged according to their classification at a given taxonomical level. Only those bacterial and archaeal taxa were included in the ALDEx2 analysis that had at least 100 sequences in total in the compared samples. The p-values from the Welch's t-tests were corrected for multiple testing with the method of Benjamini and Hochberg [41].

## Results

### Quantitative analyses (qPCR results)

Between samples from BT and non-BT maize, there was no significant difference in the abundance of bacterial 16S rRNA genes or in the abundance of *nirS* genes at any of the sites in any of the sampling years. The abundance of archaeal 16S rRNA genes was higher in the non-BT samples than in the BT samples at the Spanish site in 2012 (Table 2) but the maize genotype did not have a significant influence on the archaeal 16S rRNA gene number at the other sites in any of the sampling years and at the site in Spain in 2013. The fungal ITS sequences were more abundant in the BT samples than in the non-BT samples from the site in Sweden in 2014 (Table 2) but the maize genotype did not affect fungal ITS abundance at the other sites in any of the sampling years and at the site in Sweden in 2012 and 2013. The *nirK* copy number was higher in the non-BT samples than in the BT samples at the Slovakian site in 2012 and the Swedish site in 2014 (Table 2) but *nirK* abundance was not influenced by the maize genotype at the sites in Denmark and Spain and at the site in Slovakia in 2013 and 2014, and Sweden in 2012 and 2013.

Bacterial 16S rRNA gene numbers were in 2012 significantly higher in the samples from the Swedish site than in the samples from Slovakia and Spain (Fig 1). Between 2012 and 2013, the abundance of bacterial 16S rRNA genes significantly increased at the Slovakian and Spanish sites, but not in the samples from Sweden, and there were no differences between the four sites in 2013. Between 2013 and 2014, there was a significant decrease at all sites. Among the samples from 2014, the ones from Sweden had significantly lower bacterial 16S rRNA gene numbers than the samples from the Slovakian site.

Archaeal 16S rRNA gene abundance was in 2012 significantly lower at the site in Slovakia compared to the samples from Spain and Sweden (Fig 1). There was a significant decrease between 2012 and 2013 at the Slovakian and Swedish sites, and in 2013, the archaeal 16S rRNA gene numbers were significantly lower in the samples from Slovakia than at the other sites, and in the samples from Denmark compared to the ones from the Spanish site. Between 2013 and 2014, there was a significant decrease in the archaeal abundance at the sites in Denmark and Sweden. Among the samples from 2014, the archaeal 16S rRNA gene numbers were

**Table 2. Significant differences between MON810 maize (Bt) and isogenic comparator maize (non-Bt) cultivars in the qPCR results.**

| Site and year | Gene | Average ± SD (log copies g⁻¹ root) | | *p* |
|---|---|---|---|---|
| | | **Bt** | **non-Bt** | |
| Spain 2012 | Archaeal 16S | 7.85 ± 0.09 | 7.94 ± 0.03 | 0.0188 |
| Sweden 2014 | Fungal ITS | 7.80 ± 0.23 | 7.62 ± 0.12 | 0.0483 |
| | *nirK* | 8.98 ± 0.34 | 9.26 ± 0.09 | 0.0308 |
| Slovakia 2012 | *nirK* | 8.91 ± 0.14 | 9.13 ± 0.11 | 0.0009 |

The p-values are from Welch's t-tests

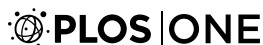

**Fig 1. Gene copies for quantification of microbial communities in maize rhizospheres (qPCR results).** Symbols indicate mean values, lines show standard deviations. Sampling years are distinguished by symbols, sites by colors. Groups not sharing the same letter within a plot are significantly different (Tukey's HSD, p < 0.05).

significantly different at the three sites being high in Slovakia, intermediate in Sweden, and low in Denmark.

For fungal ITS sequences, the numbers were significantly higher at the Swedish site than at the sites in Slovakia and Spain in 2012 (Fig 1). Between 2012 and 2013, fungal ITS abundance increased at the Slovakian and Swedish sites but did not change in Spain. In 2013, the samples from Sweden had the highest number of fungal ITS sequences followed by the samples from Denmark, Slovakia, and Spain, all being significantly different from the others. Between 2013 and 2014, the fungal ITS copy numbers decreased in the samples from Sweden, but did not change at the other sites. Thus, in 2014, the highest fungal ITS abundance was found at the site in Denmark, followed by Slovakia, and lastly Sweden, all differences being significant.

**Fig 2. Correlation between the gene copies of different target gene sequences.** Scatterplot matrices with shading from grey to red indicate low to high density of data points. The Pearson correlation coefficient is shown in the corner of the plots.

**Table 3. Genera represented by the source organisms of the reference sequences matched by the *nirK* TSVs.**

| Class | Order | Family | Genus |
|---|---|---|---|
| *Alphaproteobacteria* | *Rhizobiales* | *Bradyrhizobiaceae* | *Afipia* |
| | | | *Bradyrhizobium* |
| | | | *Rhodopseudomonas* |
| | | *Brucellaceae* | *Brucella* |
| | | | *Ochrobactrum* |
| | | *Phyllobacteriaceae* | *Chelativorans* |
| | | | *Mesorhizobium* |
| | | *Rhizobiaceae* | *Agrobacterium* |
| | | | *Neorhizobium* |
| | | | *Rhizobium* |
| | | | *Sinorhizobium* |
| | | *Xanthobacteraceae* | *Starkeya* |
| | *Rhodobacterales* | *Rhodobacteraceae* | *Phaeobacter* |
| | | | *Ruegeria* |
| *Betaproteobacteria* | *Burkholderiales* | *Alcaligenaceae* | *Achromobacter* |
| | | | *Alcaligenes* |
| *Gammaproteobacteria* | *Pseudomonadales* | *Pseudomonadaceae* | *Pseudomonas* |

In 2012, the number of *nirK* genes was significantly higher in the samples from Sweden than in the ones from Slovakia and Spain (Fig 1). From 2012 to 2013, *nirK* abundance significantly increased at all three sites. In 2013, the highest was at the sites in Denmark and Sweden followed by Spain with significantly lower copy numbers than in Sweden, and lastly Slovakia with *nirK* abundance significantly lower than the other three sites. Between 2013 and 2014, the number of *nirK* genes decreased significantly at all sites. In 2014, *nirK* abundance was significantly higher in the samples from Denmark and Sweden than at the Slovakian site.

In 2012, the copy number of *nirS* genes was higher in the samples from the Spanish site than in the samples from Slovakia and Sweden (Fig 1). From 2012 to 2013, *nirS* abundance increased significantly at the site in Sweden. In 2013, *nirS* gene numbers were the highest at the site in Spain, significantly lower in the samples from Denmark and Sweden, and lowest in the samples from Slovakia. Between 2013 and 2014, the copy number of *nirS* genes increased significantly at the site in Slovakia, but decreased in Denmark and Sweden. Accordingly, in 2014, *nirS* abundance was significantly higher in the Slovakian compared to the Danish and Swedish samples.

There was a strong positive correlation (r = 0.759) between the abundance of archaeal 16S rRNA genes and the *nirS* gene numbers (Fig 2). The *nirK* abundance was more correlated with bacterial 16S rRNA gene numbers (r = 0.525), and there was no strong correlation between *nirK* and *nirS* abundance (r = 0.207). While bacterial and archaeal 16S copy numbers were in strong correlation with each other (r = 0.691), they were not strongly correlated to fungal abundance (r = 0.390 and -0.101 respectively). All pair-wise correlations between the qPCR results were found significant (p ≤ 0.003) except for the correlation between fungal and archaeal abundance (p = 0.157).

## DNA-sequencing yields

Sequencing the V4 region of the 16S rRNA gene from the 200 samples yielded 5,660,124 high-quality bacterial and archaeal sequences grouped into 10,416 SVs with an average length of 231 bp (Supplementary file 1). The individual samples had 10,858 to 58,101 sequences and 1,079 to 4,908 SVs. *Archaea* was represented by 1.3% of the sequences. The SVs were classified

**Fig 3. Simpson diversity.** Symbols indicate mean values, lines show standard deviations. Sampling years are distinguished by symbols, sites by colors. Groups not sharing the same letter within a plot are significantly different (Tukey's HSD, p < 0.05).

into 33 bacterial and two archaeal phyla. The most abundant were *Proteobacteria* (55.9 ± 16.6%), *Actinobacteria* (23.7 ± 9.5%), *Firmicutes* (5.0 ± 2.2%), *Bacteroidetes* (3.5 ± 2.5%), *Acidobacteria* (3.3 ± 1.9%), *Chloroflexi* (2.0 ± 1.3%), *Verrucomicrobia* (1.8 ± 1.0%), and *Gemmatimonadetes* (1.7 ± 0.8). Within *Proteobacteria*, the Gamma- (31.9 ± 16.6%) and Alpha- (14.0 ± 6.1%) subphyla showed the highest relative abundance.

MiSeq *nirK* amplicon sequencing produced 2,301,204 high-quality sequences in 6,216 SVs with an average length of 414 bp. FrameBot could match 4,584 of the SVs, containing 2,014,870 sequences, to the FunGene *nirK* reference database. These SVs formed 3,014 TSVs, on average 138 amino acid residues long, which were matched to 39 reference sequences by FrameBot. The source organisms of these 39 reference sequences represent 17 genera from phylum *Proteobacteria* (Table 3).

Fungal ITS sequencing was performed from the samples from 2012 and 2013. One of the 2012 non-BT samples from Spain and one of the 2013 BT samples from Sweden had to be removed from the dataset due to very low sequencing yields. From the remaining 138 samples, 9,173,003 high-quality sequences were obtained which were grouped into 2,706 SVs. The SVs were 134 to 437 nt long with 245 nt average length. The samples contained 27,442–112,974 sequences and 38–188 SVs. The SVs had representatives of six fungal phyla: *Ascomycota* (51.7 ± 13.0%), *Basidiomycota* (27.3 ± 15.7%), *Zygomycota* (17.7 ± 7.7%), *Chytridiomycota* (1.2 ± 1.6%), *Glomeromycota* (0.050 ± 0.087%), and *Rozellomycota* (detected in two samples, 0.045% and 0.018%). 877 SVs could be classified at species-level representing 427 species and together containing 69.3% of the ITS sequences.

## Simpson diversity

The samples from BT and non-BT maize did not show significant difference in the Simpson diversity of 16S rRNA gene SVs and fungal ITS SVs at any of the sites or years. Among the samples from the Slovakian site from 2014, however, the Simpson diversity of *nirK* TSVs was significantly lower (p = 0.003) in the BT samples (0.968 ± 0.014) than in the non-BT samples (0.987 ± 0.01). In contrast, the *nirK* diversity was not influenced by the maize genotype at the other sites or at the Slovakian site in 2012 and 2013.

The Simpson index calculated from the 16S rRNA sequencing dataset showed no significant variation between sites and years with one exception: diversity was significantly lower in the Slovakian samples from 2013 than at the other sites and in other years at the site in Slovakia (Fig 3). This difference is likely a result of the high abundance of one SV (SV#1), classified as a member of *Enterobacteriaceae*, in the 2013 samples from the Slovakian site (39.1 ± 20.4%) compared to the other samples (4.2 ± 7.8%) leading to decreased evenness and thus lower diversity (Supplementary file 1).

The diversity of *nirK* TSVs was significantly higher in the samples from Sweden than at the Spanish site in 2012 (Fig 3). In 2013, the Danish and Swedish samples showed higher diversity than the ones from the Slovakian site. The year-to-year variation within the sites was not significant, except for an increase at the Slovakian site between 2013 and 2014. In 2014, there was no significant difference in *nirK* diversity between the sites.

In fungal ITS diversity, there was no significant change between 2012 and 2013 at any of the sites (Fig 3). In both years, the samples from Slovakia had significantly lower Simpson index values than the samples from the other sites.

## Community structure and composition

On the PCA plots from the 16S and ITS SVs, samples from BT plants did not separate from the non-BT samples from the same site and year, indicating that the rhizosphere prokaryotic

**Fig 4. Principal component analysis plots.** Sampling years are distinguished by symbols. Darker colors indicate samples from field plots with maize MON810 (Bt), lighter colors samples from non-modified near isogenic maize (non-Bt).

and fungal community structure was not influenced by the maize genotype. On the other hand, samples from different years form distinct groups on the plots, showing that there was considerable year-to-year variation in community structure at all sites (Fig 4A–4D and 4I–4L). Similarly, the PCA plots from the *nirK* sequencing data showed no difference between the samples from BT and non-BT plants from the same year at the Danish, Spanish, and Swedish sites, but clear separation between years among the samples from Denmark and Spain (Fig 4E, 4G and 4H). The samples from the Swedish site did not show such clear grouping by years, the samples from 2012 overlap with the ones from 2013 on the ordination plot (Fig 4H). The PCA plot from the *nirK* TSVs from the Slovakian samples indicate a clear difference in the community structure of *nirK*-type denitrifiers between the samples from 2012 and 2013 but no difference caused by the maize genotype in either of these years. The samples from 2014, however,

**Table 4. SVs and taxa from the 16S sequencing data with significantly different abundance in the rhizosphere of maize MON810 (Bt) and non-modified comparator maize (non-Bt maize).** Results from ALDEx2.

| Site and year | SV or taxon | non-Bt maize | Bt maize | *p*-value |
|---|---|---|---|---|
| Denmark 2014 | SV#198 (*Acinetobacter*) | 1.32 ± 0.39 | 0.004 ± 0.006 | 0.008 |
| | genus *Acinetobacter* | 2.74 ± 1.16 | 0.81 ± 0.55 | 0.013 |
| Slovakia 2014 | SV#5 (*Arthrobacter*) | 1.70 ± 0.33 | 0.84 ± 0.40 | 0.033 |
| | SV#20 (*Bacillales*) | 0.34 ± 0.11 | 0.14 ± 0.04 | 0.017 |
| | SV#1467 (*Sphingobium*) | 0.18 ± 0.12 | 0.004 ± 0.005 | 0.028 |
| | genus *Arthrobacter* | 1.86 ± 0.33 | 0.89 ± 0.41 | 0.031 |
| | genus *Acidothermus* | 0.044 ± 0.020 | 0.0006 ± 0.0018 | 0.047 |

The taxonomical classifications of SVs are indicated in parentheses. Relative abundances are shown as percentages (average ± standard deviation). The *p*-values are from Welch's t-tests from ALDEx2 with Benjamini-Hochberg correction.

group by maize genotype: the samples from BT plants appear close to the samples from 2012, while the samples from non-BT plants separate from all other samples from the Slovakian site (Fig 4F).

Among bacterial and archaeal SVs and taxa, the genus *Acinetobacter* (*Gammaproteobacteria*) and SV#198 classified into this genus had significantly lower relative abundance in the BT than in the non-BT rhizosphere samples at the Danish site in 2014 (Table 4). However, this genus and SV did not show significant response to the maize genotype at the other sites or in Denmark in 2013. In the samples from the Slovakian site in 2014, genera *Arthrobacter* and *Acidothermus* (both *Actinobacteria*), and three SVs assigned to *Arthrobacter*, *Bacillales* (*Firmicutes*), and *Sphingobium* (*Alphaproteobacteria*) decreased significantly in relative abundance in the BT compared to the non-BT samples (Table 4). These genera and SVs were not found to respond significantly to the maize genotype in the other two years at the site in Slovakia or at the other sites in any of the sampling years.

No fungal SVs or taxa were found to have significantly different abundance in the rhizosphere of non-BT and BT maize at any of the sites in any of the sampling years. Similarly, no *nirK* TSVs were significantly affected by maize genotype at any of the sites in any of the sampling years except at the Slovakian site in 2014. There were 161 *nirK* TSVs of significantly different abundance between the non-BT and BT rhizosphere samples from Slovakia 2014. Eighty of them were more abundant in the rhizosphere of non-BT maize and 81 in the rhizosphere of BT (S2 Table). Out of these 80 TSVs, 51 were not detectable in the samples from the BT plants. The combined relative abundance of the 161 TSVs in the Slovakian samples from 2014 was 60.2 ± 6.8%. Their nearest matching sequences from the FunGene *nirK* reference database and the source organisms of the reference sequences are listed in Table 5. Out of the 20 reference sequences, 13 had matches both among the TSVs more abundant in the non-BT rhizosphere samples and the TSVs more abundant in the samples from BT plants.

Regarding differences in community structure between the sites, similar patterns were found in the bacterial and archaeal, *nirK*-type denitrifier, and fungal communities. NMDS revealed large differences between the sites that approximate the geographic distances between them: the samples from the Swedish and Danish locations are close on the ordination plots, the Slovakian samples are further apart, while the ones from the site in Spain are relatively far from the others. The NMDS plots of the samples from 2013 are shown as an example in Fig 5. The difference in the structure of the *nirK*-type denitrifier community between the samples from non-BT and BT maize at the Slovakian site in 2014 (Fig 4F) was not large compared to the differences between sites (Fig 6).

**Table 5. Identifiers (IDs) of the nearest matching sequences from the FunGene reference database and their source organisms for the *nirK* TSVs significantly more abundant in the non-Bt or Bt samples from Slovakia 2014.**

| Reference ID | Source organism | More abundant in | |
|---|---|---|---|
| | | non-Bt | Bt |
| 1NPJ_A | *Alcaligenes faecalis* | + | |
| ACF98088 | uncultured bacterium 1042 | + | + |
| ACF98123 | uncultured bacterium 1062 | + | + |
| ACF98152 | uncultured bacterium 1116 | + | + |
| ACM24772 | *Sinorhizobium* sp. NP1 | + | + |
| NP_435927 | *Sinorhizobium meliloti* 1021 | + | |
| NP_949481 | *Rhodopseudomonas palustris* | + | + |
| Q01537 | *Neorhizobium galegae* | + | + |
| YP_001242619 | *Bradyrhizobium* sp. BTAi1 | + | + |
| YP_001314809 | *Sinorhizobium medicae* WSM419 | + | + |
| YP_002825538 | *Sinorhizobium fredii* NGR234 | + | |
| YP_004110634 | *Rhodopseudomonas palustris* | + | + |
| YP_004134222 | *Mesorhizobium ciceri* biovar *biserrulae* WSM1271 | + | + |
| YP_004614534 | *Mesorhizobium opportunistum* | + | + |
| YP_473141 | *Rhizobium etli* | + | + |
| YP_665890 | *Chelativorans* sp. BNC1 | + | |
| YP_674799 | *Chelativorans* sp. BNC1 | + | |
| ZP_03523451 | *Rhizobium etli* GR56 | + | + |
| ZP_04680002 | *Ochrobactrum intermedium* LMG 3301 | | + |
| ZP_05742967 | *Silicibacter* sp. TrichCH4B | | + |

## Discussion

Several previous studies have already assessed the effect of the genetic modification of maize MON810 on the rhizomicrobiome utilizing different techniques. Most of them failed to find an effect, and it's been assumed that there is no effect or it is not large enough to become detectable over the background of natural variation [22, 42, 43]. However, most previous studies investigated the microbial community with methods of relatively low resolution, e.g. with a widely applied gel-based genetic fingerprinting that can usually distinguish not more than 20 to 50 dominant taxonomic units [44, 45]. Higher resolution can be achieved by high-throughput DNA sequencing of PCR amplicons [10, 46]. This has revealed that a Bt-maize expressing three different Cry-genes was different in the composition of its rhizosphere microbial community from its non-GM comparator, but on the other hand, this difference was not larger than what was found between non-GM cultivars [46]. In our study however, with the sequencing depth and the number of samples representing different field sites and annual replication higher than in any of the above cited works, the results show that neither the abundance nor the diversity of markers characterizing the rhizomicrobiome was affected by maize MON810 and the expression of the Cry1Ab protein in a consistent manner.

Considering that the Cry1Ab protein has been demonstrated to enter the soil via root exudation or sloughed-off root cells [22, 47], a response of the rhizomicrobiome resulting in a differently structured community detectable by deep sequencing of marker genes would not have been surprising. The protein present in the rhizosphere could potentially interact with soil microorganisms, either acting as a toxic or growth inhibiting compound or the opposite, serving as an energy and nutrient source thereby supporting the growth of specific microbial community members. Adverse effects, including toxicity of the Cry1Ab protein on soil



**Fig 5. Non-metric multidimensional scaling plots of the samples from 2013.**

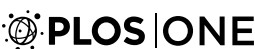

**Fig 6. Non-metric multidimensional scaling plot from the *nirK* translated sequence variants (TSVs) with the samples from 2014.**

microorganisms have been previously analyzed but never detected [48, 49]. In contrast it was demonstrated that Cry1Ab can be degraded by soil microbial cells and support microbial growth [50]. However, the stable community structure of the rhizomicrobiome found in our study suggests that the presence of Cry1Ab was not sufficient to cause significant shifts in relative abundances of microbial community members. This lack of response may be explained by the low amount of Cry1Ab compared to other plant proteins released into the soil, and also by

the fact that Cry1Ab adsorbs to surface-active soil particles which limits its accessibility to the microbiome and slows down its degradation [50, 51].

Natural variation caused by plant age, weather conditions, and soil properties have been identified as major factors which may mask Bt-specific effects on the rhizomicrobiome. To compensate for plant age, all samples in this study were collected at the same maize growth stage, i.e. during flowering. At this stage, the root system is fully developed and significant amounts of Cry1Ab protein can be detected in the rhizosphere [22]. Underlining the importance of environmental factors, distinct microbiomes were found at each site based on all three genetic markers that were analyzed: 16S rRNA gene, ITS region, and *nirK* gene. Similarly, comparing soil bacterial communities from sites across Europe, it was found that the site had a predominant influence on community structure [52].

Interestingly, the NMDS analyses indicated a higher similarity between the rhizosphere communities from the two sites from Northern Europe (Denmark and Sweden, situated 115 km apart), than between them and the other two sites, even though in Slovakia the same maize genotype was used. Apparently, the biogeographical zone was more important in shaping the rhizomicrobiome than the cultivars. This geographical effect is likely a combined result of similar soil types and climatic conditions under which maize was cultivated. Shared climate was also considered an important driving factor of bacterial diversity in the rhizosphere of maize studied with different cultivars at five field locations across the United States [53]. Apart from soil and weather, variation between the sites of this study could be linked to locally optimized agricultural management, including fertilization practices, tillage, and pest management.

The plants selected for distinct rhizomicrobiomes at each field site, still the community composition as indicated by the relative abundance of phyla, subphyla, and classes was similar to those described in other studies [53, 54]. While the self-organized process of the formation of the rhizosphere microbial community indicated by the consistent occurrence of the same taxonomic groups across different geographical zones is still a matter of research, it appears that the genetic modification did not interfere with this process. The response of vascular arbuscular (VA) mycorrhizal fungi, which commonly belong to the phylum *Glomeromycota*, to maize expressing Cry1Ab has been investigated, but Bt-specific effects found in one greenhouse study could not be confirmed under field conditions [25, 42, 55]. In our study, *Glomeromycota* was represented by only $0.050 \pm 0.087\%$ of the fungal ITS sequences, suggesting that VA mycorrhization of maize cultivated at the different field sites across Europe had no importance. It has to be noted, that PCR-based surveys of ITS diversity can underestimate the abundance of *Glomeromycota* [56, 57]. Yet, the lack of mycorrhization would not be unexpected considering that soil tillage and excessive fertilization with nitrogen, potassium, and phosphate are not unusual in maize cultivation in Europe.

The functional genes *nirK* and *nirS* were analyzed in this study because they were suspected to be sensitive to a modified rhizodeposition. In fact, bacterial communities involved in denitrification were found to respond to the quantity and quality of root exudates [58]. Several studies have indicated that, even though both genes encode an enzyme catalyzing the same reaction, their ecological patterns are different suggesting specific, yet poorly understood niche adaptation [59–62]. When comparing *nirS* and *nirK* encoded proteins, it was also suggested that *nirK* is more frequently associated with incomplete denitrification than *nirS*. Consequently, a higher expression of *nirK* versus *nirS* would lead to an increased production of the greenhouse gas $N_2O$, while higher *nirS* expression would mean larger $N_2O$ sink capacity [63, 64]. On average, *nirK* was approximately 500-fold more abundant than *nirS* in our samples. At the four sites during the three years included in our study, *nirK* abundance was affected by the genetic modification of maize only in two isolated cases and only to a minor extent. These results suggest that maize cultivation at these sites may not be favorable for avoiding the

adverse effect of $N_2O$ emission but a replacement of conventionally bred maize by Bt maize MON810 would be neutral in this regard assuming no change in agricultural management practices.

If only one year, 2014, had been included in our study, it would have concluded based on 161 differentially abundant *nirK* TSVs containing almost 70% of all *nirK* sequences, that soil bacteria with the genetic potential to reduce nitrite and to be involved in the production of the greenhouse gas $N_2O$ were affected by the genetic modification of the plant at the site in Slovakia, while at other sites this did not occur. This would suggest that the genetic modification can alter the soil microbiome depending on factors like climate or soil type. Results from the other two years from the Slovakian site, however, did not confirm this observation, showing that it was a sporadic event which did not reflect a common biological response to the cultivation of maize MON810. It should be noted that the dramatic change observed in the structure of the *nirK*-type denitrifier community in Slovakia in 2014 likely did not translate to a large effect at the functional level since many TSVs that showed the opposite response were affiliated with the same bacterial taxa. Similarly, isolated data from 2014 indicated a decline in the abundance of *nirK* genes in response to the genetic modification of maize at the Swedish site, accompanied by an increase in fungal abundance indicated by ITS sequences. Annual replication, however, did not confirm these responses, suggesting that they were owed to the high variability of microbiological parameters which may be encountered at field sites exposed to variable weather conditions. Considering the potential consequences that such misconceptions derived from data from only one season could imply for an environmental risk assessment, our study underlines the crucial importance of collecting data from more than one growing season. The importance of annual replication was also a major conclusion of a recent review on GM effects on fungal communities [65].

In conclusion, this study demonstrates, with a highly sensitive microbial community analysis which also covered less abundant community members, that the genetic modification enabling Bt maize MON810 to express the insecticidal protein Cry1Ab has no tangible effect on the composition of its rhizomicrobiome. The sporadic differences found in the abundance and community structure of rhizosphere *Bacteria*, *Archaea*, *Fungi*, and denitrifiers between MON810 maize and its near-isogenic comparators were minor or unlikely to affect community functions. Furthermore, these were isolated cases not consistently observed in subsequent years. Our results clearly indicate that for an accurate assessment of the effects of GM plants on their rhizomicrobiome, annual replication is even more important than the consideration of field sites from different biogeographical zones.

## Supporting information

**S1 File. Data matrices obtained from high-throughput sequencing.**
(XLSX)

**S1 Table. Sequences of primers and probes used in qPCR.**
(DOCX)

**S2 Table. The nirK translated sequence variants (TSVs) differentially abundant between the non-BT and BT samples from the Slovakian site in 2014.** Relative abundances are indicated as percentages (average ± standard deviation). The p-values are from Welch's t-tests from ALDEx2 with Benjamini-Hochberg correction. The IDs of the nearest matching sequences from the FunGene *nirK* reference database and their source organisms are included.
(DOCX)

## Acknowledgments

We thank all members of the EU project AMIGA for collaboration, especially Salvatore Arpaia (ENEA, Rotondella) and Antoine Messéan (INRA, Grignon-Paris) for coordination and discussion. We would also like to thank our colleagues running the field experiments for their hospitality and strong support during our visits, especially Cristina Chueca (INIA, Madrid), Tina D'Hertdfeldt (Lund University), Gábor L. Lövei (Aarhus University, Slagelse), Ľudovít Cagáň (Slovak Agricultural University), and to all members of their respective teams. We also thank Jana Usarek, Karin Trescher, and Britta Müller for their excellent technical assistance and Sinead Phelan and Vilma Ortiz Cortés in Teagasc. The study was funded by the Seventh Framework Program of the EU, Grant agreement 289706 (AMIGA: Assessing and Monitoring the Impacts of Genetically modified plants on Agro-ecosystems, paper No. 40).

## Author Contributions

**Conceptualization:** Márton Szoboszlay, Christoph C. Tebbe.

**Data curation:** Márton Szoboszlay, Astrid Näther, Christoph C. Tebbe.

**Formal analysis:** Márton Szoboszlay, Astrid Näther.

**Funding acquisition:** Christoph C. Tebbe.

**Investigation:** Márton Szoboszlay, Astrid Näther, Ewen Mullins, Christoph C. Tebbe.

**Methodology:** Márton Szoboszlay, Astrid Näther, Ewen Mullins, Christoph C. Tebbe.

**Project administration:** Christoph C. Tebbe.

**Resources:** Márton Szoboszlay, Ewen Mullins, Christoph C. Tebbe.

**Software:** Márton Szoboszlay.

**Supervision:** Christoph C. Tebbe.

**Validation:** Astrid Näther, Ewen Mullins, Christoph C. Tebbe.

**Visualization:** Márton Szoboszlay, Astrid Näther, Christoph C. Tebbe.

**Writing – original draft:** Márton Szoboszlay, Christoph C. Tebbe.

**Writing – review & editing:** Márton Szoboszlay, Astrid Näther, Ewen Mullins, Christoph C. Tebbe.

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
