## [Decision Letter · Decision Letter 0]

14 Oct 2019

PONE-D-19-24482

Annual replication is essential in evaluating the response of the soil microbiome to the genetic modification of maize in different biogeographical regions

PLOS ONE

Dear Dr. Christoph Tebbe,

Thank you for submitting your manuscript to PLOS ONE. After careful consideration, we feel that it has merit but does not fully meet PLOS ONE’s publication criteria as it currently stands. Therefore, we invite you to submit a revised version of the manuscript that addresses the points raised during the review process.

We would appreciate receiving your revised manuscript by Nov 28 2019 11:59PM. To enhance the reproducibility of your results, we recommend that if applicable you deposit your laboratory protocols in protocols.io, where a protocol can be assigned its own identifier (DOI) such that it can be cited independently in the future. For instructions see: http://journals.plos.org/plosone/s/submission-guidelines#loc-laboratory-protocols

We look forward to receiving your revised manuscript.

Kind regards,

Luigimaria Borruso

Academic Editor

PLOS ONE

Journal Requirements:

Additional Editor Comments (if provided):

Reviewers' comments:

Reviewer's Responses to Questions

**Comments to the Author**

1. Is the manuscript technically sound, and do the data support the conclusions?

Reviewer #1: Partly

Reviewer #2: Yes

2. Has the statistical analysis been performed appropriately and rigorously? 

Reviewer #1: Yes

Reviewer #2: Yes

3. Have the authors made all data underlying the findings in their manuscript fully available?

Reviewer #1: Yes

Reviewer #2: Yes

4. Is the manuscript presented in an intelligible fashion and written in standard English?

Reviewer #1: Yes

Reviewer #2: Yes

5. Review Comments to the Author

Reviewer #1: The paper "Annual replication is essential in evaluating the response of the soil microbiome to the

genetic modification of maize in different biogeographical regions" is interesting and intriguing paper, but my first impression is that, despite a large amount of data it should have been obtained from analyses, there is a is little in-depth analysis and enhancement of them. I think that it should be enhanced, by re-organizing it in a more linear structure. The Introduction is very clear and specific and it well focus on the subject of the work. Furthermore, it is also very smooth and pleasant to read. Materials and methods are very detailed but not very fluent, so I suggest to change the text to make it more pleasant, perhaps by eliminating repeating parts. The Results and discussion should be ordered, them are dispersive, it would be good to reorganize them in a more schematic way. Discussions are interesting and well focused on the results, but need to be reworked and more detailed.

I provide some suggestions for the authors, as follows:

114-121: it would be good to insert the data of the soils in a table (or insert them in table 1). Moreover, have the chemical-physical data been collected every year? Or just once? It is not clear, please clarify it.

122-130: is the protocol already described in some work and/or modified by some other paper on the same topic? please specify it.

178-180: why authors used two libraries with different chemistries performed? Since the aim of the work is a comparative study of the microbiome, this could lead to an error in the comparison of the results. Did authors consider it? Please state this choice.

204: why was the 123 database chosen (release 2015), considering that the most recent version is 132 ( release 2018)?

322-329: I suggest to outline the community found in a table, with the percentage of phyla and genus retrieved in samples. I would also add a table with sequencing yields results reported in the text (easier to follow).

410-417: this sentence is very convoluted, it would be good to write the concept better.

430-432: the results are interesting, but why authors showed only the graph related to one of the sampling years? Authors should show them all in a single graph.

462-472: was the concentration of the Cry1Ab samples measured? If yes, authors should add it to material and methods as well as in the results. If not, all that period should be modified.

Reviewer #2: Number: PONE-D-19-24482

Title: Annual replication is essential in evaluating the response of the soil microbiome to the genetic modification of maize in different biogeographical regions

General comments

The research is original, following established scientific procedures and it is scientifically sound. The manuscript is well structured and the writing is concise and the English level allows a fluent lecture to understand the concepts and the authors’ intent. Therefore, please clarify a few things and finally, I would like to recommend the manuscript for the publication in PLOS ONE.

Detailed comments

Line 39: Instead of colon (:), I would rather prefer a point (.) after “decades”.

Line 40: Please write Nowadays instead of “Today”.

Line 68: Please use a comma (,) instead of “and” after “18S rRNA”.

Line 71: Please write Among them instead of “E.g”.

Line 73: Please add the abbreviation (N) after “nitrogen”.

Line 117- 121: Please write “…total C and …total N contents”.

Line 131: Please uniform the modus of reporting Throbäck et al. (2004) according the Journal’s requests.

Line 251-253: Although the results in terms of abundance of 16S rRNA and nirS genes were not significant as a function of site and sampling year, I would like to see them in Table 2. Please add them.

Line 370-372: I am wondering why the results of ITS diversity for the year 2014 are not shown in Figure 3. Please double-check.

Figure 5: Why did you choose to perform the NMDS analysis of results concerning the year 2013 and not for the other two years (2012 and 2014)?

References: Please check and uniform the references following the author's guidelines.

Line 599: Please write “Bacillus thuringiensis” in italic font.

6. PLOS authors have the option to publish the peer review history of their article (what does this mean?). If published, this will include your full peer review and any attached files.

Reviewer #1: No

Reviewer #2: No

---

## [Author Response · Author response to Decision Letter 0]

24 Oct 2019

Dear Editor and Reviewers,

we are grateful for your constructive comments that helped us improved the manuscript. Below we address the points and questions raised.

Reviewer #1: 

R:The paper "Annual replication is essential in evaluating the response of the soil microbiome to the genetic modification of maize in different biogeographical regions" is interesting and intriguing paper, but my first impression is that, despite a large amount of data it should have been obtained from analyses, there is a is little in-depth analysis and enhancement of them. I think that it should be enhanced, by re-organizing it in a more linear structure. The Introduction is very clear and specific and it well focus on the subject of the work. Furthermore, it is also very smooth and pleasant to read. Materials and methods are very detailed but not very fluent, so I suggest to change the text to make it more pleasant, perhaps by eliminating repeating parts. The Results and discussion should be ordered, them are dispersive, it would be good to reorganize them in a more schematic way. Discussions are interesting and well focused on the results, but need to be reworked and more detailed.

A:This work was based on a large amount of data which, we agree, could be used to support the investigation of further questions. The aim of this manuscript, however, was not to present every bit of information that is possible to gain from the data but to fulfill the specific goals of the project as outlined in the introduction and present a focused study. The Results and Discussion sections are structured in a linear fashion along these goals. The materials and methods section was checked for redundancy. The data can be utilized to explore further questions in future manuscripts and is made available to the scientific community both as the raw sequencing output (European Nucleotide Archive) and a data matrices of the SV and TSV counts (Supplementary file 1).

R:I provide some suggestions for the authors, as follows:

114-121: it would be good to insert the data of the soils in a table (or insert them in table 1). Moreover, have the chemical-physical data been collected every year? Or just once? It is not clear, please clarify it.

A: The soil parameters were measured at every sampling time but had little variation. The goal of the project was not to study the effects of soil parameters on the rhizomicrobiome of maize, thus they are shown only as background information and were not used in the data analysis. Since the four sites were located in different biogeographical regions, they differed not just in soil parameters. It was more appropriate to use “site” in the analysis as a factor that combines the effects of soil type, climate etc.

R: 122-130: is the protocol already described in some work and/or modified by some other paper on the same topic? please specify it.

A: Obtaining samples of the rhizosphere by washing closely attached soil particles off the roots is a common practice. Yet the exact sample masses, volumes, and centrifugation parameters are optimized to each case and thus differ between studies.

R: 178-180: why authors used two libraries with different chemistries performed? Since the aim of the work is a comparative study of the microbiome, this could lead to an error in the comparison of the results. Did authors consider it? Please state this choice.

A: Sequencing of the bacterial and archaeal 16S rRNA amplicons from the samples from 2012 and 2013 took place before the sampling in 2014. At that time, the V3 chemistry was not available. Sequencing the 16S rRNA gene from the 2014 samples, and the nirK and ITS from all of the samples were performed later with the V3 chemistry. We agree that the potential bias of the different sequencing chemistries can add to or modify the differences in the 16S data between the 2014 samples and the samples from the other two years. Unfortunately, this effect can’t be separated from the year-to-year variation in this case. Being aware of this limitation, we did not quantify or test the significance of the differences in bacterial and archaeal community structure between 2014 and the other two years. The only exception is the Simpson diversity index, which we compared between the years. In this case, however, the only significant difference we found was driven by the high abundance of one SV in the Slovakian samples from 2013 (lines 356 – 361). We consider it extremely unlikely that this result is caused by the different sequencing chemistries. The analysis of the bacterial and archaeal community structure was focused on comparing the Bt to the non-Bt samples from the same site and year which were processed with the same sequencing chemistry in each case. A year-to-year variation in the bacterial and archaeal community structure was observed on the PCA plots (Figure 4 A-D). Potential biases of the sequencing chemistries can contribute to, but certainly not solely responsible for the distance of the 2014 samples from the other samples on the plots, especially considering the clear separation between the 2012 and 2013 samples. Accordingly, the presence of year-to-year variation was stated but it´s effect size was not quantified and responding SVs or taxa were not reported (lines 377 – 380).

R: 204: why was the 123 database chosen (release 2015), considering that the most recent version is 132 (release 2018)?

A: Version 123 was the most up-to-date release of the SILVA reference at the time of the analysis of the 16S sequencing data.

R: 322-329: I suggest to outline the community found in a table, with the percentage of phyla and genus retrieved in samples. I would also add a table with sequencing yields results reported in the text (easier to follow).

A: Such an outline of the community in a table format would result in two enormous tables, one for 16S the other for ITS, with 556 and 427 rows respectively. Furthermore, it would provide much information relevant to the objectives of the manuscript. Interested readers can easily generate these tables from the SV matrices provided in Supplementary file 1.

R: 410-417: this sentence is very convoluted, it would be good to write the concept better.

A: This section consists of short sentences with simple structure. They are not describing a concept just making simple statements.

R: 430-432: the results are interesting, but why authors showed only the graph related to one of the sampling years? Authors should show them all in a single graph.

A: The site in Denmark was not sampled in 2012 and the site in Spain in 2014 (Table 1 and also line 123). Therefore, the data from 2013 is the best to demonstrate the point raised in this section as it includes all the sites.

R: 462-472: was the concentration of the Cry1Ab samples measured? If yes, authors should add it to material and methods as well as in the results. If not, all that period should be modified.

A: The concentration of the Cry1Ab protein was not measured in the rhizosphere samples. The fact that MON810 releases it in its rhizosphere has been demonstrated previously in several studies (see references in the section).

Reviewer #2:

General comments

The research is original, following established scientific procedures and it is scientifically sound. The manuscript is well structured and the writing is concise and the English level allows a fluent lecture to understand the concepts and the authors’ intent. Therefore, please clarify a few things and finally, I would like to recommend the manuscript for the publication in PLOS ONE.

Detailed comments

R: Line 39: Instead of colon (:), I would rather prefer a point (.) after “decades”. - A: Corrected.

R: Line 40: Please write Nowadays instead of “Today”. A:Corrected.

R: Line 68: Please use a comma (,) instead of “and” after “18S rRNA”. A:“And” is appropriate.

R: Line 71: Please write Among them instead of “E.g”. A: Corrected.

R: Line 73: Please add the abbreviation (N) after “nitrogen”. A: We don’t consider it necessary to indicate the chemical symbol of nitrogen. It’s known by the readers.

R: Line 117- 121: Please write “…total C and …total N contents”. A: Corrected.

R: Line 169: Please uniform the modus of reporting Throbäck et al. (2004) according the Journal’s requests. A:The references were updated.

R: Line 251-253: Although the results in terms of abundance of 16S rRNA and nirS genes were not significant as a function of site and sampling year, I would like to see them in Table 2. Please add them.

A: Differences between sites and sampling years are shown in Figure 1.

R: Line 370-372: I am wondering why the results of ITS diversity for the year 2014 are not shown in Figure 3. Please double-check.

A: The samples from 2014 were not included in the fungal ITS sequencing due to financial constraints. We apologize that a clear statement on this was missing from the method section. We added it to lines 181 and 341.

R: Figure 5: Why did you choose to perform the NMDS analysis of results concerning the year 2013 and not for the other two years (2012 and 2014)?

A: The NMDS plots in Figure 5 serve to demonstrate that regarding differences between the sites, a similar pattern was found in the 16S, nirK, and ITS datasets approximating the geographical distances between the sites. The site in Denmark was not sampled in 2012 and the site in Spain in 2014 (Table 1 and also line 123). Therefore, the data from 2013 is the best to demonstrate this as it includes all the sites.

R: References: Please check and uniform the references following the author's guidelines.

A: The references were updated.

R: Line 599: Please write “Bacillus thuringiensis” in italic font. A: Corrected.

---

## [Decision Letter · Decision Letter 1]

26 Nov 2019

PONE-D-19-24482R1

Annual replication is essential in evaluating the response of the soil microbiome to the genetic modification of maize in different biogeographical regions

PLOS ONE

Dear Dear Prof. Dr. Christoph C Tebbe,

Thank you for submitting your manuscript to PLOS ONE. After careful consideration, we feel that it has merit but does not fully meet PLOS ONE’s publication criteria as it currently stands. Therefore, we invite you to submit a revised version of the manuscript that addresses the points raised during the review process.

I will be delighted to accept personally your manuscript after you have responded to the last few requirements of Reviewer 2.

We would appreciate receiving your revised manuscript by Jan 10 2020 11:59PM. To enhance the reproducibility of your results, we recommend that if applicable you deposit your laboratory protocols in protocols.io, where a protocol can be assigned its own identifier (DOI) such that it can be cited independently in the future. For instructions see: http://journals.plos.org/plosone/s/submission-guidelines#loc-laboratory-protocols

We look forward to receiving your revised manuscript.

Kind regards,

Luigimaria Borruso

Academic Editor

PLOS ONE

Reviewers' comments:

Reviewer's Responses to Questions

**Comments to the Author**

1. If the authors have adequately addressed your comments raised in a previous round of review and you feel that this manuscript is now acceptable for publication, you may indicate that here to bypass the “Comments to the Author” section, enter your conflict of interest statement in the “Confidential to Editor” section, and submit your "Accept" recommendation.

Reviewer #1: All comments have been addressed

Reviewer #2: (No Response)

2. Is the manuscript technically sound, and do the data support the conclusions?

Reviewer #1: Yes

Reviewer #2: Yes

3. Has the statistical analysis been performed appropriately and rigorously? 

Reviewer #1: Yes

Reviewer #2: Yes

4. Have the authors made all data underlying the findings in their manuscript fully available?

Reviewer #1: Yes

Reviewer #2: Yes

5. Is the manuscript presented in an intelligible fashion and written in standard English?

Reviewer #1: Yes

Reviewer #2: Yes

6. Review Comments to the Author

Reviewer #1: All comments indicated in the review have been addressed in a clear and categorical way. I am fully satisfied.

Reviewer #2: Number: PONE-D-19-24482R1

Title: Annual replication is essential in evaluating the response of the soil microbiome to the genetic modification of maize in different biogeographical regions

General comments

The quality of the revised manuscript has been improved. Thanks to the authors that have followed the reviewers’ comments. Therefore, please clarify a few things and finally, I would like to recommend the revised manuscript for the publication in PLOS ONE.

Detailed comments

Line 122: Please remove a bracket.

Line 379: “did not” instead of “do not”.

Line 384: “showed” instead of “show.

Line 386: “did not” instead of “do not”.

7. PLOS authors have the option to publish the peer review history of their article (what does this mean?). If published, this will include your full peer review and any attached files.

Reviewer #1: No

Reviewer #2: No

---

## [Author Response · Author response to Decision Letter 1]

26 Nov 2019

All four modifications suggested for the 2nd revision have been accepted and implemented

---

## [Editor Report · Decision Letter 2]

2 Dec 2019

Annual replication is essential in evaluating the response of the soil microbiome to the genetic modification of maize in different biogeographical regions

PONE-D-19-24482R2

Dear Dr. Christoph C Tebbe,

We are pleased to inform you that your manuscript has been judged scientifically suitable for publication and will be formally accepted for publication once it complies with all outstanding technical requirements.

With kind regards,

Luigimaria Borruso

Academic Editor

PLOS ONE
---

## [Editor Report · Acceptance letter]

6 Dec 2019

PONE-D-19-24482R2 

Annual replication is essential in evaluating the response of the soil microbiome to the genetic modification of maize in different biogeographical regions 

Dear Dr. Tebbe:

I am pleased to inform you that your manuscript has been deemed suitable for publication in PLOS ONE. Congratulations! Your manuscript is now with our production department. 

With kind regards,

on behalf of

Dr. Luigimaria Borruso 

Academic Editor

PLOS ONE